# Bullying in Middle School: Evidence for a Multidimensional Structure and Measurement Invariance across Gender

**DOI:** 10.3390/children10050873

**Published:** 2023-05-12

**Authors:** Georgios Sideridis, Mohammed H. Alghamdi

**Affiliations:** 1Boston Children’s Hospital, Harvard Medical School, Boston, MA 02115, USA; 2Department of Primary Education, National and Kapodistrian University of Athens, 10679 Athens, Greece; 3Department of Self Development Skills, King Saud University, P.O. Box 2454, Riyadh 11451, Saudi Arabia; mhalghamdi@ksu.edu.sa

**Keywords:** bullying, TIMSS, elementary school, alignment, construct validity, gender differences

## Abstract

The purpose of the present study was to evaluate the factorial structure of the bullying scale on the Trends in International Mathematics and Science (TIMSS 2019) for eighth graders and evaluate the instrument’s invariance across gender so that tests of level between males and females can be conducted. Data came from the 2019 cohort of TIMSS in Saudi Arabia. The 14-item scale was evaluated using three competing models: (a) a unidimensional structure, (b) the International Association for the Evaluation of Educational Achievement (IEA) online, non-online two-factor model, and (c) the Wang et al. (2012) 4-domain bullying taxonomy. Participants were 5567 eighth graders who participated in the 2019 TIMSS study. There were 2856 females and 2711 males. The mean age was 13.9 years. Data were analyzed using Confirmatory Factor Analysis (CFA) and Mplus 8.9. Results indicated that a 4-domain structure including verbal, physical, relational, and online bullying represented the most optimal factor structure of the 14-item bullying measure. Tests of exact measurement invariance for gender originally failed but were then satisfied using the newly recommended “alignment” methodology. Latent mean differences were salient and significant suggesting that levels of bullying across all domains were elevated in males compared to females, contrasting earlier views that different types of bullying are linked to males versus females. Results are discussed in relation to educational policy interventions.

## 1. Introduction

According to the National Center for Education Statistics (2019), one in five students in the United States has been bullied at some point in their education. Various studies have shown percentages ranging from 8.6% to 45.2%, and the World Health Organization (WHO) estimates that this proportion is approximately 33% worldwide (World Health Organization, 2019) [1,2].

Several studies have been conducted to investigate the prevalence of bullying as well as its effects, including its connections to mental health and academic outcomes [3,4]. For instance, ref. [5] conducted a meta-analysis and discovered that victims of bullying were at a higher risk of developing depression, anxiety, and suicidal ideation (see also [6,7]). On the other hand, those who bully others were more likely to exhibit conduct problems and substance use. According to the findings of another study carried out by [8], being a victim of bullying was associated with lower academic achievement as well as higher absenteeism [9].

A variety of prevention and intervention strategies have been implemented in schools and communities across the country to combat bullying [10,11,12]. These strategies include school-wide campaigns, social-emotional learning programs, and peer mediation programs. According to the findings of several studies, these programs are effective in lowering instances of bullying behavior and enhancing students’ social and emotional well-being [13,14]. Nevertheless, putting into action efficient strategies for prevention and intervention can be difficult, and there is still a significant amount of work to be done to address bullying in schools and communities. Parents, teachers, and policymakers need to work together and coordinate their efforts to accomplish this goal.

### 1.1. Challenges in the Measurement of Bullying at School

There are many reasons why bullying may not be identified when it occurs. One of the reasons is that it is not always reported, which can happen either because the person who is being bullied is too afraid to speak up or because they are unaware that they are the target of bullying [15]. This indicates that the true extent of bullying may not be accurately reflected in the literature that is currently available. In addition, bullying can take many different forms, including physical, verbal, and emotional abuse, which can make it difficult to recognize and measure, particularly given the lack of established definitions for the term [16]. Last but not least, self-report instruments are the methodological tool that is used the most frequently, even though there is a possibility that issues such as anonymity, social desirability, response bias, acquiescence, and recall bias may be at play, which hinders the true responses of participants [17,18,19,20].

### 1.2. Measurement of Bullying Internationally: The TIMSS and PIRLS 2019 Instrument

Undoubtedly, the evaluation of the prevalence of bullying is tied to the validity of those measurements [21]. Consequently, issues of reliability and validity are of primary importance. Table 1 displays the content of the bullying scale in the TIMSS 2019 context, which consists of 14 items anchored between a 1 and 4 scaling system (ranging from “never” to “at least once a week”). Currently, empirical evidence on the measurement of bullying using TIMSS has pointed to the presence of either a unidimensional construct [22] or a two-factor correlated domain assessing online and non-online bullying [23] (IEA instrument, Table 1, second column from the right). However, an examination of the instrument’s content using the eighth-grade cohort data suggests that items may tap on additional domains. For example, [24] suggested that there are four major forms of bullying, physical, verbal, relational, and cyber. Thus, the well-known physical behaviors (e.g., hitting, pushing), and verbal (e.g., calling names), are supplemented by the more complex forms of relational bullying (e.g., social exclusion, rumor spreading) and the more recent forms involving the internet and new technologies (termed online or cyberbullying, see [25,26,27,28,29,30].

### 1.3. Gender Differences in Bullying Behaviors at School

There is ample evidence that levels and types of bullying are moderated by gender [11,26,31,32,33,34,35]. Research has demonstrated that there are distinct gender variations in both the frequency of bullying and the forms that it takes. According to the findings of two extensive meta-analytic studies [24,36], boys have a greater propensity to take on the role of the bully than girls do. These findings, however, are qualified by the form of bullying that took place. According to several studies [26,31,32,33,37,38,39], boys are more likely to be the victims of both traditional and online forms of bullying. On the other hand, relational bullying, which is manipulating relationships to injure the victim, appears to be more prevalent in females [24,36]. This type of bullying is defined as “harming the victim through manipulation of relationships”. Although some studies have supported the interaction between types of bullying and gender, several explanations have been proposed [40]. For example, socialization models appear to differ between males and females as males may be seen positively when being assertive, dominant, and competitive whereas girls may be viewed positively when they appear nurturing and empathetic. Consequently, there are expectations that females may be more prone to relational bullying compared to males [36,41].

### 1.4. The Problem and Importance of the Present Study

The current investigation sought to provide a more thorough comprehension of bullying in the context of TIMSS. By scrutinizing simplified structures and incorporating new domains, we can disprove earlier studies and develop a more accurate method for measuring bullying and harassment at school. This will help us shift from a one-dimensional view of bullying to a multidimensional one that better depicts the phenomenon’s complexity. In addition, the investigation aims to assess the measurement parity of abuse across genders. Instead of assuming that males and females experience similar levels of abuse, we seek to provide empirical evidence that either supports or refutes this assumption. Thus, we can make valid comparisons between male and female bullying levels, which can inform policies and interventions aimed at reducing bullying in schools. Overall, the current investigation’s objectives are essential for advancing our understanding of bullying in the context of the TIMSS. By providing a more accurate and comprehensive measurement of abuse and examining its equivalence across genders, we can more effectively address this crucial issue and promote a safer and more inclusive learning environment for all students.

## 2. Method

### 2.1. Participants and Procedures

The participants in this study were 5680 eighth graders from Saudi Arabia who took part in the TIMSS 2019 study. Only participants who provided complete data were considered for inclusion. There were 2791 males (49.2%), and 2884 females (50.8%, data on gender were missing from 5 participants). The students come from 224 different schools located in different parts of Saudi Arabia. Additional details on the study’s methodology can be located at https://timssandpirls.bc.edu/timss2019 (accessed on 1 May 2023).

### 2.2. Instrument: Bullying Scale of the TIMSS 2019 Measurement

It was assessed using a 14-item scale as shown in Table 1. The items are as follows: frequency of (1) mean things being said about me (2), lies being spread about me (3), secrets being shared with others (4), refused to talk (5), my family being insulted (6), something being stolen from me (7), forced to do something (8), sent nasty messages (9), shared things online (10), shared photos online, (11) threatened me (12), hurt me (13), excluded me, and (14) damaged something belonging to me. The items were classified as belonging to four bullying types, namely, physical, verbal, relational, and cyber (see Table 1) based on earlier work [24]. The scaling system involved a frequentist four-point scaling as follows: Never, A few times a year, Once or twice a month, and At least once a week. A reversed coding procedure was implemented so that higher scores would be indicative of higher levels of bullying.

### 2.3. Data Analyses

#### 2.3.1. Internal Consistency Reliability

Internal consistency reliability was assessed using Cronbach’s alpha [42] despite shortcomings on inflation [43], given that those coefficients that originate in the Confirmatory Factor Analysis model such as maximal and omega [44] internal consistency reliability cannot be estimated with domains having only two items [44], as models are not identified (non-positive degrees of freedom). Estimates ranged between 0.60 and 0.80; specifically, verbal bullying had an alpha of 0.803, relational bullying 0.634, physical bullying 0.683, and online bullying 0.727. Notably, these coefficients represent low-bound estimates of internal consistency reliability.

#### 2.3.2. Confirmatory Factor Analysis

Confirmatory factor analysis, also known as CFA, is a multivariate approach that examines the relationships between a set of observed items or indicators to determine whether or not a theoretical construct or latent variable may be confirmed as being genuine [45,46,47]. It is a method of structural equation modeling that entails specifying and evaluating a model with a priori hypothesized relationships among the latent variable and its indicators [48]. Comparing the observed data, which are typically presented in the form of a variance-covariance matrix, to the values that the model predicts for that matrix is the objective of CFA. This will help determine whether or not the hypothesized model provides a good fit for the data. Goodness-of-fit indices like chi-square, Comparative Fit Index (CFI), Tucker–Lewis Index (TLI), and Root Mean Square Error of Approximation (RMSEA) is frequently used to evaluate how well a model fits its data. Estimates of descriptive fit indices that are greater than 0.900 or 0.950 are suggestive of good model fit [49,50]. The respective estimate for unstandardized residuals (RMSEA) ranges between 5 and 8 percentage points [45] as indicators of “close” fit with estimates less than 5% being indicative of “exact” fit. Last, the chi-square test is given less precedence given its sensitivity to enhanced power and the emittance of Type-I errors.

#### 2.3.3. Measurement Invariance: Exact Protocol

Measurement invariance refers to the degree to which a certain instrument produces the same results across a variety of populations [51,52]. More specifically, the concept of measurement invariance is an assertion that the scores and interpretations of a measure are the same for all of the groups [53]. It is a crucial presumption because, without it, it is impossible to determine whether observed differences are the result of real population differences or variations in the fundamental concept that is being evaluated [54]. Without this presumption, it is impossible to distinguish between the two possibilities. If it is not there, the scores that are obtained from measurement cannot be compared between different groups [55]. For point-estimate comparisons, the equality of component structures (also known as configural invariance), item slopes (often known as metric invariance), and item intercepts/thresholds (commonly known as scalar invariance) are the three levels of measurement invariance that are required [56]. Recent developments have suggested using approximate invariance protocols or the alignment technique to achieve measurement invariance, in addition to having exact protocols of measurement invariance, for which the Confirmatory Factor Analysis model is used [57]. This would be in addition to having exact protocols of measurement invariance [58]. If the exact measurement invariance protocol does not hold, the alignment methodology that is described in the next section will be utilized in this study [58] as a means of satisfying partial measurement invariance and still estimating potential differences between latent means across gender.

#### 2.3.4. Measurement Invariance: Alignment Method

Using a two-step process, the alignment methodology developed by [58] is a method that can be implemented in structural equation modeling as a strategy for achieving measurement invariance across different groups. First, an independent baseline model is estimated for each group independently and without any constraints. The estimates of the factor loadings and intercepts/thresholds are then constrained between the different groups in the second step. The alignment methodology allows for partial invariance, which means that some of the factor loadings or intercepts/thresholds can be freely estimated, while others are constrained to be equal across groups or time points. This allows for a more accurate representation of the relationship between the factors. This approach has the potential to be especially helpful in situations in which there are significant reasons to anticipate substantial differences in the factor structure or item loadings between groups.

We opted for the alignment protocol compared to approximate measurement invariance for the following reasons: With approximated MI, the model looks for a solution where the measurement parameters’ variance is minimal. Nonetheless, the objective of the suggested alignment approach is to identify a solution that incorporates measurement parameters “with a substantial degree of minor non-invariance” (p. 7). Because there are no additional limitations placed on the configural model because of the alignment methods, the model tries to maximize invariance by allowing factor means and variances to vary across groups. The fact that differences between groups are still conceivable even if exact measurement invariance protocols are unsuccessful is one of its primary advantages for employing the alignment protocol. All analyses were conducted using Mplus 8.9 [59]. Within the model then that invariance is satisfied, and estimated latent means are contrasted. Within Mplus, the latent mean(s) of the first (reference) group is fixed to zero and the estimated latent mean of the focal group reflects the difference between reference and focal means.

## 3. Results

### 3.1. Factorial Validity of Bullying Scale

A series of CFA models were employed to identify the optimal factor structure of the bullying scale of the TIMSS. Those involved two unidimensional models, (one with fixed and one with free slopes, as in mimicking the Rasch model in Item response models), a 2-factor correlated model as specified by IEA, and last a 4-factor correlated model put forth by Smith et al. These results are shown in Table 2.

The freely estimated slopes unidimenionsal model provided improved fit to the data compared to the fixed slopes unidimensional model as it was associated with significant improvement in model fit by almost 23k chi-square units. Furthermore, the two-factor model posited by IEA suggested that online and non-online dimensions described the data better compared to the unidimensional structure. The cost of ignoring the online dimension was approximately 251 chi-square units. Last, the four-domain correlated model was tested positing four theoretically proposed domains, namely, verbal, relational, physical, and online/cyber. This last model provided the best fit to the data with an additional improvement in model fit by 110 chi-square units. Consequently, the four-domain solution provided the most parsimonious measurement of the latent construct of bullying. The descriptive fit indices were excellent (CFI = 0.982, TLI = 0.977), and unstandardized residuals were less than 5% (i.e., 0.046) pointing to “exact fit” [60] of the data to the model.

### 3.2. Measurement Invariance across Gender for Four-Factor Bullying Model

One of the important prerequisites to conducting point-estimate differences across groups is the assumption that the instrument operates in the same way across different populations or occasions. To this end, the classic model of “exact” measurement invariance was first examined. Based on this model, the necessary steps involve testing the equivalence of simple structures (configural invariance), followed by the additional imposition of equal item slopes (metric invariance), and, last, followed by the imposition that the intercepts/thresholds of the indicators are also equivalent across groups (scalar invariance). Table 3 shows model fit statistics from testing “exact” measurement invariance.

As shown in the table, despite the overall good model fit as suggested by descriptive fit indices and the RMSEA, there was a significant difference between configural and metric models [ΔChi-square(10) = 136.067, *p* < 0.001] and between the metric and scalar models [ΔChi-square(10) = 163.815, *p* < 0.001]. Consequently, the exact measurement invariance protocol failed and thus, the alignment methodology was employed next.

The alignment model was run using maximum likelihood estimation with robust standard errors (MLR). The fixed alignment method was utilized that fixed the factor variances of the first group (males) to unity and the factor means across all four domains to zero. Table 4 displays the final solution along with tests of thresholds and slope invariance. The last two columns on the right indicate the equivalence of slope and threshold terms across gender (as there were four choices in the scaling system there were three thresholds). As shown in the table, there was only one threshold for which non-invariance was evident, i.e., the three-dimensional threshold of item BSBG14M, which is an item related to the frequency of exclusion by peers.

### 3.3. Gender Differences in Bullying at the Latent Mean Level

Figure 1 displays differences between the bullying constructs across gender after achieving invariance using alignment. As shown in the figure, there were significant differences across gender on all four bullying constructs. Specifically, there were significantly lower levels of bullying for females across both verbal and online/cyberbullying with the larger effect sizes approaching medium levels after rounding (i.e., 0.5 standard deviation [61]). Third in order of magnitude came relational bullying with medium to large effects and last, with a small effect size came physical bullying, all denoting higher levels for males compared to females.

## 4. Discussion

The purpose of the present study was to evaluate the factorial structure of the bullying scale on the TIMSS 2019 for eighth graders and evaluate the instrument’s invariance across gender so that tests of level between males and females can be conducted. Several important findings emerged and are discussed next.

The most significant finding was that there was no connection between the type of bullying and a person’s gender as boys continually engaged in bullying behavior at significantly higher rates compared to girls. These findings disagree with past research in which types of bullying exerted specialized effects across gender [24,26,37,62,63,64,65]. Additionally, these findings are home to a plethora of different concepts and ideas. To begin, the earlier result that social and cultural variables explain why females participate in relational bullying to a larger degree than males was not confirmed in the present study [39]. On the contrary, we found that males participate in relational bullying at a higher level compared to females. The foundation of this stereotype is the idea that men are more likely to engage in more overt forms of aggression, such as punching and causing damage to property, whereas women are more likely to engage in more covert forms of aggression as a means of imposing their control over others and resolving conflicts [34]. The above idea is further supported by the premise that according to society’s standards, physical aggression in males is perceived to be more acceptable than it is in females, and it is even reinforced as a gender quality [66].

The second most important finding relates to providing a more complete framework under which bullying is being measured and evaluated internationally. The examination of a four intercorrelated domain framework found support from the data in TIMSS and suggests that future investigations will continue to engage this conceptualization compared to either a unidimensional, or two-domain construct structure. This proposition, however, is valid only if the present four-factor model is supported and replicated by other populations, thus, tests of measurement invariance across countries on the TIMSS are necessary. In other words, the present simple structure should not be taken for granted but be evaluated, verified, or disproved with future samples.

Third, despite notable differences in the functioning of the bullying scale across gender, the application of the novel alignment methodology [58] overcame the problem of construct equivalence and allowed us to perform valid tests in levels between males and females. Thus, the present study acknowledged recent advances in psychometrics that overcame the problem of measurement non-equivalence [53].

### 4.1. Implications for Educational Policy and Practice

The results of the present study allow for several recommendations that can influence educational policy. First, it is recommended that schools involve parents, stakeholders, principals, and specialists (such as psychologists and social workers) in addressing bullying because of the intricacy of this problematic conduct. It is recommended that regulations be put in place to detect and treat toxic masculinity, which is a term that represents society’s acceptance that boys may and must be aggressive, domineering, and macho, in light of the findings of the current study on male dominance. Training programs that emphasize the use of empathy in conflict resolution and the maintenance and development of healthy social interactions in males are likely to have a favorable impact. Additionally, the development of clear guidelines on what constitutes bullying, harassment, and aggressive instances will help recognize such incidents, thus providing the first step of recognition and assessment. Enhancing accountabilities for all students so that incidents will be reported and receive proper consequences may also create a culture of zero tolerance toward bullying.

### 4.2. Limitations and Future Directions

There are several reasons why the present findings should be viewed with caution. First, the self-reported nature of the measurement suggests that processes such as response bias and social desirability may be operative [67], especially at that young age. Second, the sample size was relatively large so significant effects may not have been “substantial” but rather an artifact of excessive levels of power. For example, the difference in cyberbullying across gender was reflective of small effect sizes, pointing to that effect. Last, given the use of the TIMSS methodology and the specific instrument, it is possible that the content validity of the measure can be improved by incorporating additional behaviors related to cyber and online means of bullying. Such behaviors can now be assessed in younger ages as there has been an increase in bullying prevalence in younger ages.

### 4.3. Conclusions

It is concluded that the measurement of bullying in the TIMMS is better described using a multivariate framework that is richer, more explanatory, and offers the potential for targeted interventions. Furthermore, gender differences were noted favoring females who exhibited lower levels of bullying throughout the bullying domains.

## Figures and Tables

**Figure 1 children-10-00873-f001:**
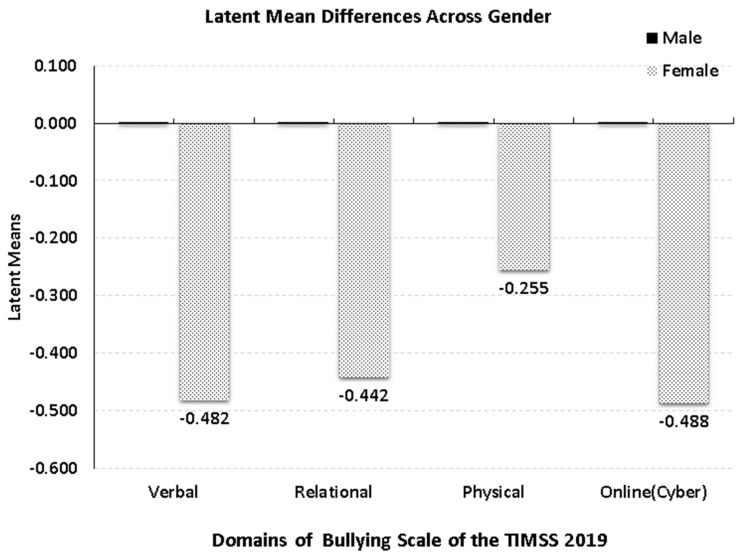
Latent mean differences in bullying across gender using factor scores from the 4-factor correlated model using the alignment method. Mean estimates for males were fixed to zero with the estimates shown reflecting deviations in bullying levels for females compared to males.

**Table 1 children-10-00873-t001:** Items and Domains for Bullying Scale of TIMSS 2019 in Grade 8.

Item No.	TIMSS 2019 Item Label	Item Content	Bullying Domains IEA	Bullying Domains Smith et al., 2019 [24]
1	BSBG14A	How often mean things were said about me	Non-Online	Verbal
2	BSBG14B	How often lies were spread about me	Non-Online	Verbal
3	BSBG14C	How often secrets were shared with others	Non-Online	Verbal
4	BSBG14D	How often I refused to talk	Non-Online	Relational
5	BSBG14E	How often my family was insulted	Non-Online	Verbal
6	BSBG14F	How often I was stolen from	Non-Online	Physical
7	BSBG14G	How often I have been forced to do something	Non-Online	Physical
8	BSBG14H	How often nasty messages were sent to me	Online/Cyber	Online/Cyber
9	BSBG14I	How often things were shared online about me	Online/Cyber	Online/Cyber
10	BSBG14J	How often photos of me were shared online	Online/Cyber	Online/Cyber
11	BSBG14K	How often I was threatened	Non-Online	Verbal
12	BSBG14L	How often I was hurt	Non-Online	Physical/Verbal
13	BSBG14M	How often I was excluded	Non-Online	Relational
14	BSBG14N	How often someone damaged something belonging to me	Non-Online	Physical

**Table 2 children-10-00873-t002:** Model fit of bullying scale of the TIMSS 2019.

Model	Chi-Square	D.F.	Model Comparison	ΔChi-Square	ΔD.F.	*p*-Value
M1. Fixed slopes Unidimensional	24,051.070	91	-	-	-	-
M2. Free slopes Unidimensional [39]	1267.757	77	M1 vs. M2	22,783.313 ***	14	<0.001
M3. 2-Factor IEA	1016.363	76	M2 vs. M3	251.394 ***	1	<0.001
M4. 4-Factor Smith et al.	906.210	71	M3 vs. M4	110.153 ***	5	<0.001

Note: Uni = Unidimensional; D.F. = Degrees of Freedom; ΔChi-square = Difference chi-square estimate between nested models; ΔD.F. = Difference in degrees of freedom between nested models; *p*-value = reflects the likelihood that the chi-square test is zero. *** *p* < 0.001.

**Table 3 children-10-00873-t003:** Exact measurement invariance of bullying scale of the TIMSS 2019.

Model Tested	Chi-Square	D.F.	*p*-Value	CFI	TLI	RMSEA
M1. Configural Model	1773.300 ***	142	<0.001	0.938	0.920	0.064
M2. Metric Model	1909.368 ***	152	<0.001	0.933	0.920	0.064
M3. Scalar Model	2073.183 ***	162	<0.001	0.927	0.918	0.065

Note: D.F. = Degrees of freedom; CFI = Comparative fit index; TLI = Tucker–Lewis index; RMSEA = Root mean squared error of approximation.*** *p* < 0.001.

**Table 4 children-10-00873-t004:** Estimates of intercepts and slopes across gender using alignment.

Items	Males Item Slopes	Females Item Slopes	Males Item Thresholds	Females Item Thresholds	Aligned Slopes	Aligned Thresholds
Verbal Bullying						
BSBG14A	1.748	1.688	−3.608/−2.603/−1.532	−2.997/−2.069/−1.178	Yes	Yes/Yes/Yes
BSBG14B	1.976	1.764	−3.879/−2.669/−1.055	−3.737/−2.498/−1.471	Yes	Yes/Yes/Yes
BSBG14C	1.749	1.835	−3.420/−2.434/−1.313	−4.025/−2.990/−2.029	Yes	Yes/Yes/Yes
BSBG14E	2.504	2.371	−5.379/−4.514/−3.372	−5.068/−3.951/−2.969	Yes	Yes/Yes/Yes
BSBG14K	3.037	3.161	−7.326/−6.113/−4.666	−7.508/−6.092/−4.839	Yes	Yes/Yes/Yes
BSBG14L	2.375	2.695	−5.842/−4.869/−3.454	−6.215/−4.923/−3.657	Yes	Yes/Yes/Yes
Relational Bullying						
BSBG14D	2.311	2.221	−4.933/−3.945/−2.581	−4.465/−3.388/−2.514	Yes	Yes/Yes/Yes
BSBG14M	2.798	2.911	−6.116/−4.915/−3.317	−6.184/−5.155/−3.930	Yes	Yes/Yes/No
Physical Bullying						
BSBG14F	1.335	1.591	−3.623/−2.639/−1.414	−3.991/−2.954/−1.765	Yes	Yes/Yes/Yes
BSBG14G	2.294	2.265	−5.679/−4.750/−3.511	−5.613/−4.450/−3.658	Yes	Yes/Yes/Yes
BSBG14N	2.512	2.363	−5.649/4.589/−3.066	−5.514/−4.318/−2.829	Yes	Yes/Yes/Yes
Online/Cyber Bullying						
BSBG14A	1.973	1.965	−4.332/−3.429/−2.342	−4.097/−3.291/−2.513	Yes	Yes/Yes/Yes
BSBG14B	2.987	2.759	−6.268/−5.302/−4.119	−5.863/−5.024/−4.112	Yes	Yes/Yes/Yes
BSBG14C	2.499	3.079	−6.608/−5.748/−4.758	−7.266/−5.995/−4.907	Yes	Yes/Yes/Yes

## Data Availability

Data are available from the official study of TIMSS 2019.

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
