# Peer review of "Bullying in Middle School: Evidence for a Multidimensional Structure and Measurement Invariance across Gender"

_children, 2023, doi:10.3390/children10050873_

Round 1

Reviewer 1 Report

Dear Authors,

You present an interesting and still relevant problem of bullying.

The submitted abstract is clear and complete.

The introduction of the article is clear, the methods section is presented in detail, as well as the results section.

I would suggest paying special attention to the discussion part. 

I would suggest that you compare your results with other authors' data and explain them in more detail. Maybe there are studies with results close to yours.

I would also recommend a separate section of conclusions.

Author Response

Thank you for your thoughtful comments. Below please find our responses to your queries: 

Dear Authors,

You present an interesting and still relevant problem of bullying.

Answer:

Thank you, very much appreciated.

The submitted abstract is clear and complete.

Answer:

Thank you, very much appreciated.

The introduction of the article is clear, the methods section is presented in detail, as well as the results section.

Answer:

Thank you, very much appreciated.

I would suggest paying special attention to the discussion part. 

Answer:

Thank you.

I would suggest that you compare your results with other authors' data and explain them in more detail. Maybe there are studies with results close to yours.

Answer:

Thank you, we have now enriched our discussion section to include more relevant studies and also add to the extant literature on gender differences in bullying.

I would also recommend a separate section of conclusions.

Answer:

Thank you, we have now added a section on conclusions.

Reviewer 2 Report

I was pleased to read the manuscript entitled "Bullying in Middle School: Evidence for a Multidimensional Structure" and to review it.

The study examined the factorial structure of a several school bulling scale and evaluated the instrument's invariance across gender. It also provide a theoretical framework and practical guidance for optimizing the assessment of bulling. From a scientific point of view, it was interesting to read the article as it is one of the few studies that was conducted in the selected population and using modern methods of analysis. The article is written in a typical format and is well structured. In my opinion, the content of the article is not fundamentally flawed, but some corrections may be appropriate.

Title – the title does not accurately reflect the manuscript. It should emphasize 'invariance of instrument across gender'. For example, it could be: Measuring Bulling in School: Evidence for Multidimensional Approach across Gender.

Abstract – some conclusion is needed. It is necessary to explain the abbreviations.

Introduction – OK. Introduction provide sufficient theoretical background for the study. All examined research questions and/or hypotheses were introduced The introduction is structured logically and the text is fluent. The rationale of the study is well described and the study problem is stated clearly. Relevant and unbiased literature was used. It is necessary to explain the abbreviations.

Method – sampling and measures are shortly described and are appropriate to answer the proposed research questions. Here I recommend to explain how the latent mean differences were calculated. In lines 110-114, select the capital or lowercase letters to use in the list.

Results – in general, results are clearly organized and presented. However, several inaccuracies were noted:

Discussion – the structure of the Discussion is clear. The interpretations is appropriate and is supported by the results. The study findings are discussed with relevant literature. However, the conclusions are needed.

References – #16, #53 and #62 are incomplete. #45 and #46 specify the publisher

Thank you for considering my opinion. I encourage authors to keep on working to improve the manuscript.

Minor editing of English language required.

Author Response

Thank you for your thoughtful comments, below please find how we responded to each one of your queries:

I was pleased to read the manuscript entitled "Bullying in Middle School: Evidence for a Multidimensional Structure" and to review it.

Answer:

Thank you, very much appreciated.

The study examined the factorial structure of a several school bulling scale and evaluated the instrument's invariance across gender. It also provide a theoretical framework and practical guidance for optimizing the assessment of bulling. From a scientific point of view, it was interesting to read the article as it is one of the few studies that was conducted in the selected population and using modern methods of analysis. The article is written in a typical format and is well structured. In my opinion, the content of the article is not fundamentally flawed, but some corrections may be appropriate.

Title – the title does not accurately reflect the manuscript. It should emphasize 'invariance of instrument across gender'. For example, it could be: Measuring Bulling in School: Evidence for Multidimensional Approach across Gender.

Answer:

Thank you, we have now changed the title to emphasize your proposition on invariance. It now reads as follows: “Bullying in middle school: Evidence for a multidimensional structure and measurement invariance across gender.”

Abstract – some conclusion is needed. It is necessary to explain the abbreviations.

Answer:

Thank you, we have now added a conclusions section.

Introduction – OK. Introduction provide sufficient theoretical background for the study. All examined research questions and/or hypotheses were introduced The introduction is structured logically and the text is fluent. The rationale of the study is well described and the study problem is stated clearly. Relevant and unbiased literature was used. It is necessary to explain the abbreviations.

Answer:

Thank you, we have now explained all abbreviations.

Method – sampling and measures are shortly described and are appropriate to answer the proposed research questions. Here I recommend to explain how the latent mean differences were calculated. In lines 110-114, select the capital or lowercase letters to use in the list.

Answer:

Thank you, we have added some text on the latent mean differences that reads as follows: “Within the model then that invariance is satisfied, estimated latent means are contrasted. Within Mplus, the latent mean(s) of the first (reference) group is fixed to zero and the estimated latent mean of the focal group reflects the difference between reference and focal means.”

Please note that estimation of latent means is a very complex procedure in which “optimal” scores are created from item intercepts using maximum likelihood. The process of latent mean estimation has been described in detail in the works of Brown (2015) and Klein (2015).

Results – in general, results are clearly organized and presented. However, several inaccuracies were noted:

Discussion – the structure of the Discussion is clear. The interpretations is appropriate and is supported by the results. The study findings are discussed with relevant literature. However, the conclusions are needed.

Answer:

Thank you, we have now added a conclusions section.

References – #16, #53 and #62 are incomplete. #45 and #46 specify the publisher

Answer:

Thank you, we have added the relevant information.

Thank you for considering my opinion. I encourage authors to keep on working to improve the manuscript.

Answer:

Thank you.

Round 2

Reviewer 1 Report

Dear authors, I hope that my observations were useful to you.